# Impact of immunosuppressive therapy on SARS-CoV-2 mRNA vaccine effectiveness in patients with immune-mediated inflammatory diseases: a Danish nationwide cohort study

Rahma Elmahdi [1,2] Daniel Ward [1] Martin T Ernst,[3] Gry Poulsen [1]
Jesper Hallas [3] Anton Pottegård [3] Tine Jess [1,2]

RE and DW are joint first authors.

¹Department of Clinical Medicine, Aalborg Universitet, Copenhagen, Denmark
²Department of Gastroenterology and Hepatology, Aalborg University Hospital, Aalborg, North Denmark Region, Denmark
³Clinical Pharmacology and Pharmacy, Department of Public Health, University of Southern Denmark, Odense, Syddanmark, Denmark

**Correspondence to**
Dr Rahma Elmahdi;
rahmae@dcm.aau.dk

## ABSTRACT

**Objective** Patients receiving immunosuppressives have been excluded from trials for SARS-CoV-2 vaccine efficacy. Investigation of immunosuppressants' impact on effectiveness of vaccines, particularly in patients with immune-mediated inflammatory diseases (IMID), is therefore required.

**Design** We performed a nationwide cohort study to assess the risk of COVID-19 infection in vaccinated patients with IMID exposed to immunosuppressives compared with IMID unexposed to immunosuppressives. Exposure to immunosuppressives in the 120 days before receiving the second SARS-CoV-2 mRNA vaccination was assessed. Patients were followed from date of second vaccination and weighted Cox models were used to estimate the risk of infection associated with immunosuppressives. Secondary outcomes included hospitalisation and death associated with a positive SARS-CoV-2 test. Risk of infection by immunosuppressant drug class was also analysed.

**Setting** This study used population-representative data from Danish national health registries in the period from 1 January to 30 November 2021.

**Results** Overall, 152 440 patients were followed over 19 341 person years. Immunosuppressants were associated with a significantly increased risk of infection across IMID (HR: 1.4, 95% CI 1.2 to 1.5), in inflammatory bowel disease (IBD) (HR: 1.6, 95% CI 1.4 to 1.9) and arthropathy (HR: 1.3, 95% CI 1.1 to 1.4) but not psoriasis (HR: 1.1, 95% CI 0.9 to 1.4). Immunosuppressants were also associated with an increased risk of hospitalisation across IMID (HR: 1.4, 95% CI 1.1 to 2.0), particularly in IBD (HR: 2.1, 95% CI 1.0 to 4.1). No significantly increased risk of death in immunosuppressant exposed patients was identified. Analyses by immunosuppressant drug class showed increased COVID-19 infection and hospitalisation with anti-tumour necrosis factor (TNF), systemic corticosteroid, and rituximab and other immunosuppressants in vaccinated patients with IMID.

**Conclusion** Immunosuppressive therapies reduced effectiveness of mRNA SARS-CoV-2 vaccination against infection and hospitalisation in patients with IMID. Anti-TNF, systemic corticosteroids, and rituximab and other

## STRENGTHS AND LIMITATIONS OF THIS STUDY

⇒ Use of a non-selected, population representative cohort to source patients with immune-mediated inflammatory disease (IMID).
⇒ Inclusion of a total of 184 346 immunosuppressive exposed patients with IMID and 152 440 propensity score matched, unexposed controls.
⇒ Complete vaccination, and immunosuppressive treatment exposure data along with complete infection, hospitalisation and death outcome data with no loss to follow-up.
⇒ Lack of individual-level data on level of exposure to infection, such as shielding behaviour.

immunosuppressants were particularly associated with these risks.

## INTRODUCTION

SARS-CoV-2 mRNA vaccines Comirnaty (Pfizer-BioNTech) and Spikevax (Moderna) were found to be efficacious in clinical trials prior to authorisation, and by December 2022 over 758 million doses of Pfizer-BioNTech and 164 million doses of Moderna were administered in the European Union.[1] Premarketing trials excluded individuals considered at risk of immunocompromise, including those receiving immunosuppressive therapies[2 3]; therefore, there remains a paucity of data on the real-world effectiveness of SARS-CoV-2 vaccines in patients treated with immunosuppressive drugs.

Research published early in the SARS-CoV-2 epidemic suggested that, in the absence of vaccination, some types of immunosuppressives including rituximab, sulfasalazine and corticosteroids are associated with an increased risk of severe outcomes in COVID-19 infection.[4–8] Immune-mediated

inflammatory diseases (IMID), including inflammatory bowel disease (IBD), inflammatory arthropathy and psoriasis, have themselves independently been associated with lower serological responses to SARS-CoV-2 vaccination than in healthy controls.[9] Immunosuppressants are key therapies in IMID, so patients with IMID may be at increased risk of infection and severe outcomes of COVID-19 infection both due to the natural history of the diseases and the therapies used to treat them. Even in the context of second vaccination against SARS-CoV-2, exposure to immunosuppressives has been associated with a significantly poorer humoral response, lower than that which is required to confer immunity against infection and severe outcomes of COVID-19 infection in patients treated with immunosuppressive therapies.[10–12]

It is therefore important to investigate the impact of immunosuppressants on SARS-CoV-2 vaccine effectiveness, while controlling for the underlying disease-indicating treatment, and other confounders that may impact vaccine effectiveness.

The real-world effectiveness of SARS-CoV-2 mRNA vaccinations against COVID-19 infection and associated outcomes such as hospitalisation or death, in the context of immunosuppressive therapy exposure among patients with IMID, has not yet been investigated. The aim of this study was to use Danish nationwide population-based data to assess the impact of immunosuppressive exposure on the risk of COVID-19 infection in three cohorts of vaccinated patients with IMID.

## MATERIALS AND METHODS
### Data sources
We conducted a nationwide cohort study using the Danish COVID-19 cohort,[13] based on data from the Danish Microbiology Registry,[14] which includes individual-level information on vaccine type, dose and date of administration; SARS-CoV-2 test type and date administered. These data were linked at the individual level to both the Danish National Patient Registry[15] and the Danish National Prescription Registry[16] using a unique Danish Civil Registration number (assigned to all individuals residing in Denmark). The Danish National Patient Registry, a register of hospital activities, includes medical diagnoses coded using International Classification of Disease (ICD-10), and medical procedures and prescriptions including treatment with intravenous medications. The Danish National Prescription Registry contains information on prescriptions dispensed at all community retail pharmacies, including date of dispensing, tablet strength and pack sizes. Ethics board review is not required for epidemiological research using nationwide registers in Denmark as data are pseudonymised and does not involve patients.

### Population, follow-up and outcomes
The IMID cohort comprised three disease-specific cohorts of all patients diagnosed with IBD, including Crohn's disease and ulcerative colitis (ICD-10: K50, K51), inflammatory arthropathy, including ankylosing spondylitis, other inflammatory spondylopathies, seropositive rheumatoid arthritis, other rheumatoid arthritis and psoriatic and enteropathic arthropathy (ICD-10: M45, M46, M05, M06, M07) or psoriasis (ICD-10: L40) in Denmark, who had received two doses of SARS-CoV-2 mRNA (Pfizer-BioNTech or Moderna) vaccine. Exclusion criteria were not receiving two doses of SARS-CoV-2 mRNA vaccine and migration prior to receipt of second vaccination. Patients with more than one of these IMID diagnoses were included in only one cohort, with IBD taking precedence, then inflammatory arthropathy, finally psoriasis. Therefore, only patients with psoriasis and neither an IBD nor an inflammatory arthropathy diagnosis were included in the psoriasis cohort. This order was preferred as extent of organ-specific disease which likely determines the dose for immunosuppressive therapy. Registration of IMID is based on clinical diagnoses, in line with national and international guidelines, such as European Crohn's and Colitis Organisation and European Society of Gastrointestinal and Abdominal Radiology (ECCO-ESGAR) guidelines for IBD diagnosis.[17 18]

Patients were followed from the date of administration of second mRNA vaccine (the index date) after 1 January 2021. The primary outcome was a positive SARS-CoV-2 PCR test in the observation period. Secondary outcomes were hospitalisation between 7 days prior to and up to 28 days after a positive test or death within 60 days of a positive test (both recorded in Danish National Patient Register). Follow-up was censored at administration of a third vaccination, emigration, death (in the absence of a positive SARS-CoV-2 test) or the end of the study period, 30 November 2021 (figure 1), as prevalence of the omicron variant became substantial after 28 November 2021.[19] As the registers are complete for the presence of patients up to emigration or death, all patients are retained until the event and there are no missing data.

### Patient and public involvement
None.

### Exposures
The exposures for this study include dispensed prescriptions or hospital administration of an immunosuppressive in the 120 days preceding the index date (date of administration of second vaccination). Immunosuppressants included selective immunosuppressants, anti-tumour necrosis factors (TNFs), interleukin inhibitors, calcineurin inhibitors, corticosteroids, rituximab and other immunosuppressants (see online supplemental table 1 for complete list and Anatomical Therapeutic Chemical classification codes for immunosuppressants). The 120-day exposure window is chosen to cover the largest pack sizes of prescriptions which can contain medications for up to 120 days. A minimum daily dose of corticosteroids equivalent to 7.5 mg prednisolone per day was estimated as the entire dispensed quantity of corticosteroids during

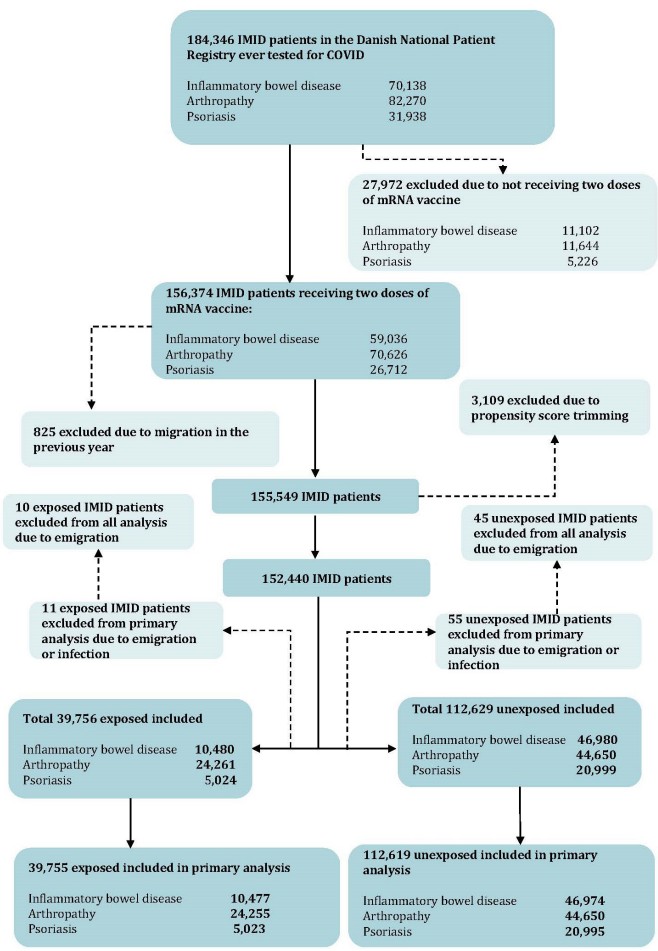

**Figure 1** Flow chart for inclusion into the exposed and unexposed immune-mediated inflammatory diseases (IMID) cohort.

a sequence of prescriptions (within the 120-day exposure period) divided by the number of days from the first prescription to the index date (online supplemental table 2). Unexposed patients with IMID were defined as those with a diagnosis of one of the three IMID, who had not received an immunosuppressive in the 120 days preceding the index date, and those receiving <7.5 mg prednisolone-equivalent average per day.

### Statistical models
Covariates included age (handled as a continuous covariate), sex, comorbidities (cardiovascular disease, pulmonary disease, liver disease, kidney disease, other gastrointestinal diseases, skin disease and musculoskeletal disease) and medications (cardiovascular drugs, antibiotics, oral anticoagulants, diabetes and chronic airway disease medications). Testing frequency varied during the period studied due to changes in national and international guidelines and travel restrictions, along with the background prevalence of SARS-CoV-2, which could introduce bias in case detection. We therefore adjusted for individual testing frequency by including number of tests in the month preceding index date as a continuous covariate. Further, covariates specific to each IMID were included separately

for each cohort. For the IBD cohort, this included any IBD-related hospital admissions in the previous year, Crohn's disease, ulcerative colitis, 5-acetylsalicylic acid/sulfasalazine, budesonide, IBD-related procedures and endoscopy of the gastrointestinal tract (see online supplemental table 3 for complete list of IMID cohort specific covariates).

To balance the covariates in the exposed and unexposed groups, we fitted propensity score (PS) models for each IMID cohort separately. PSs were calculated using logistic regression for the probability of exposure (treatment with immunosuppressives) conditional on the covariates defined above.[20] We subsequently implemented the PS using standardised mortality ratio (SMR) weights (with trimming of subjects with extreme weights beyond 1st and 99th centiles). We assessed the distribution of covariates with standardised differences before and after PS weighting.

We used weighted Cox proportional hazards regression models[21] to estimate risk of the COVID-19 infection in patients with IMID exposed to immunosuppressive therapy compared with unexposed patients for each disease cohort separately. We used calendar time as the underlying time scale to account for period effects on the risk of the outcomes which may relate to varying infection prevalence and patient characteristics as patients vulnerable to severe outcomes were vaccinated earlier in the year.

We performed secondary analyses to further delineate the impact of immunosuppressives on vaccine effectiveness over time by stratifying time since vaccination into the following intervals: 0–3 months, 3–6 months and 6–11 months. This not only allowed us to capture the period effects of COVID-19 infection risk earlier and later in the pandemic period but also allowed us to assess the impact of censoring at different time points in the follow-up period.

We then undertook Fixed Effects Model meta-analysis to calculate the pooled HR of infection, hospitalisation, and death for IBD, arthropathy and psoriasis cohorts as overall risk in immunosuppressive exposed IMID by COVID-19 outcome, and the HR of infection during 0–3, 3–6 and 6–11 months of follow-up period, as overall risk in immunosuppressive exposed IMID by period.

Finally, we also undertook drug-specific analysis for risk of COVID-19 infection by immunosuppressive drug class. In this analysis, patients receiving multiple immunosuppressive treatments were treated as independently exposed to each drug class. To account for the potential impact of immunosuppressants commonly prescribed in a weaning dose, which would not be captured using the definition of ≥7.5 mg dose equivalent per day, we undertook a sensitivity analysis to assess whether having any prescription for systemic corticosteroids over the 120-day period before the index date had an impact on the risk of infection, hospitalisation or death for those exposed to this class of immunosuppressants.

## RESULTS

A total of 184 346 patients diagnosed with IBD, arthropathy or psoriasis were identified. After exclusion of patients not receiving two doses of SARS-CoV-2 mRNA vaccine, migration prior to receipt of second vaccination and trimming of those with extreme PSs, a total 152 440 patients were included, contributing a total 19 341 person years of follow-up. During the 120-day exposure assessment period, 39 765 patients with IMID received immunosuppressive treatment (10 480 IBD, 24 261 arthropathy and 5023 psoriasis), and 112 629 patients with IMID (46 980 IBD, 44 650 arthropathy and 201 999 psoriasis) did not. A total of 11 exposed and 55 unexposed patients with IMID are censored from overall analysis due to migration or inclusion on the date of study end (therefore contributing no follow-up time). One patient with arthropathy in the exposed group, and 10 patients with IBD or psoriasis in the unexposed group are excluded from the infection analysis due to positive test on the date of study entry; these are subsequently included in the analysis for risk of hospitalisation or death following COVID-19 infection (figure 1; online supplemental table 4). Following application of SMR weighting, the cohorts were balanced on the included covariates (see table 1 for covariate prevalence and standardised differences).

A total of 866 (2.2%) COVID-19 infections were recorded among immunosuppressive exposed patients with IMID during the follow-up period compared with 2077 (1.8%) for unexposed patients with IMID. This gave an incidence rate of 55 (49–61) per 1000 person years in immunosuppressive exposed patients with IBD compared with 43 (40–45) in unexposed patients with IBD, 40 (37–44) per 1000 person years for immunosuppressive exposed patients with arthropathy compared with 38 (35–41) in unexposed patients with arthropathy, and 45 (37–55) per 1000 person years for immunosuppressive exposed patients with psoriasis compared with 42 (38–46) per 1000 person years in unexposed patients with psoriasis.

A significantly increased weighted hazard for infection among exposed patients was seen for both IBD (HR: 1.6, 95% CI 1.4 to 1.9) and arthropathy (HR: 1.3, 95% CI 1.1 to 1.4) cohorts but not for the psoriasis cohort (HR: 1.1, 95% CI 0.88 to 1.4). Meta-analysis of the three IMID cohorts showed a pooled HR for COVID-19 infection in exposed patients of 1.4 (95% CI 1.2 to 1.5; figure 2). Fewer than 57 exposed and 122 unexposed patients with IMID were hospitalised with COVID-19 infection during the follow-up period, which corresponded to a significantly increased risk of hospitalisation overall for immunosuppressive exposed patients with IMID (pooled HR: 1.4, 95% CI 1.0 to 2.0). This increased risk of overall hospitalisation is largely due to the contribution of the risk of hospitalisation in patients with IBD (2.05; 95% CI 1.03 to 4.07). Less than five immunosuppressive exposed patients with IBD died in the 60 days following a COVID-19 diagnosis compared with six unexposed patients with IBD. Six patients in the arthropathy cohort exposed to immunosuppressives compared with 13 unexposed patients with arthropathy died in the 60 days following a COVID-19 diagnosis and fewer than 5 patients with a psoriasis diagnosis, either immunosuppressive exposed or unexposed, died with a COVID-19 diagnosis. These did not correspond to a significantly increased risk of death among exposed patients in any of the three cohorts or overall (pooled HR: 0.92, 95% CI 0.38 to 2.2; figure 2).

In the first 0–3 months following vaccination, immunosuppressive exposed patients with both IBD and arthropathy had a significantly increased risk of COVID-19 infection (HR: 1.5, 95% CI 1.1 to 2.2 and HR: 1.3, 95% CI 1.0 to 1.7, respectively; see online supplemental table 5). Most COVID-19 infections following second vaccination occurred in the 3–6 months period with a total of 470 infections in exposed patients with IMID compared with 1241 unexposed patients with IMID. Only exposed patients with arthropathy had a significantly increased risk of infection compared with their unexposed counterparts during this period however (HR: 1.2, 95% CI 1.1 to 1.4). The highest incidence rate of COVID-19 infection following second vaccination was seen in the 6–11 months period for both immunosuppressive exposed and unexposed patients with IMID and risk of infection during this period was only increased among exposed patients with IBD (HR: 1.4, 95% CI 1.1 to 1.9). There was however a high rate of censoring among both the immunosuppressive exposed (over 50%) and the unexposed (almost 49%) groups in the 6–11 months period due to receipt of the third SARS-CoV-2 vaccination so direct comparison of the risk of infection between time periods is challenging. Kaplan-Meier plots and HR showing probability of infection over the calendar time of follow-up (January–November 2021) are presented in online supplemental figure 1 and post-hoc analysis for HR for infection by calendar period (January–November 2021) is shown in online supplemental table 6.

Analysis for risk of infection among IMID cohorts by immunosuppressive drug class exposure showed a significantly increased risk of infection among users of anti-TNF (HR: 1.8, 95% CI 1.6 to 2.0), systemic corticosteroid (HR: 1.2, 95% CI 1.0 to 1.5), and rituximab and other immunosuppressant (HR: 1.3, 95% CI 1.1 to 1.4; figure 3). No other immunosuppressant was significantly associated with COVID-19 infection following second vaccination. Anti-TNF and systemic corticosteroid exposure were also associated with an increased risk of COVID-19-associated hospitalisation (HR: 1.8, 95% CI 1.0 to 3.3 and HR: 1.8, 95% CI 1.0 to 3.0, respectively; figure 4). No immunosuppressive drug class was associated with death among patients with IMID following receipt of second vaccination (online supplemental table 7). Sensitivity analysis assessing outcomes following any systemic corticosteroid exposure in the 120-day period prior to the index date showed no significant difference in risk of infection (crude HR: 1.2, 95% CI 1.0 to 1.4; adjusted HR: 1.1, 95% CI 0.95 to 1.3) or death (crude HR: 3.1, 95% CI 1.4 to 7.0; adjusted HR: 1.9, 95% CI 0.77 to 4.7). However, risk

**Table 1** Characteristics of patients with immune-mediated inflammatory disease (IMID) at baseline and after propensity score weighting, by exposure to immunosuppressive therapy

| | Baseline IMID cohort | | | Weighted IMID cohort | | |
|---|---|---|---|---|---|---|
| | Unexposed | Exposed | SD | Unexposed | Exposed | SD |
| Total, n (%) | 112 675 (100) | 39 765 (100) | NA | 39 524 (100) | 39 765 (100) | NA |
| Inflammatory bowel disease (IBD), n (%) | 47 001 (41.7) | 10 480 (26.4) | NA | 10 284 (26.0) | 10 480 (26.4) | NA |
| Arthropathy, n (%) | 44 669 (39.6) | 24 261 (61.0) | NA | 24 227 (61.3) | 24 261 (61.0) | NA |
| Psoriasis, n (%) | 21 005 (18.6) | 5024 (12.6) | NA | 5014 (12.7) | 5024 (12.6) | NA |
| Age, median (IQR) | 59 (46–71) | 58 (45–71) | 0.06 | 58 (44–71) | 58 (45–71) | 0.01 |
| Male, n (%) | 48 941 (43.4) | 17 080 (43.0) | 0.01 | 16 934 (42.8) | 17 080 (43.0) | 0.00 |
| SARS-CoV-2 test in the previous month, median (IQR) | 0 (0–1) | 0 (0–1) | 0.04 | 0 (0–1) | 0 (0–1) | 0.00 |
| Calendar date of entry 2021, n (%) | | | | | | |
| January–April | 27 865 (24.7) | 16 689 (42.0) | 0.37 | 10 124 (25.6) | 16 689 (42.0) | 0.35 |
| May–August | 81 682 (72.5) | 22 361 (56.2) | 0.34 | 28 211 (71.4) | 22 361 (56.2) | 0.32 |
| September–November | 3128 (2.8) | 715 (1.8) | 0.07 | 1189 (3.0) | 715 (1.8) | 0.08 |
| Comorbidities, n (%) | | | | | | |
| Cardiovascular disease | 41 056 (36.4) | 14 010 (35.2) | 0.03 | 14 050 (35.5) | 14 010 (35.2) | 0.01 |
| Pulmonary disease | 14 994 (13.3) | 5793 (14.6) | 0.04 | 5796 (14.7) | 5793 (14.6) | 0.00 |
| Liver disease | 3630 (3.2) | 1528 (3.8) | 0.03 | 1525 (3.9) | 1528 (3.8) | 0.00 |
| Kidney disease | 6859 (6.1) | 2260 (5.7) | 0.02 | 2279 (5.8) | 2260 (5.7) | 0.00 |
| Other gastrointestinal diseases | 21 505 (19.1) | 7086 (17.8) | 0.03 | 6984 (17.7) | 7086 (17.8) | 0.00 |
| Skin disease | 9889 (8.8) | 3748 (9.4) | 0.02 | 3703 (9.4) | 3748 (9.4) | 0.00 |
| Musculoskeletal disease | 45 393 (40.3) | 16 620 (41.8) | 0.03 | 16 649 (42.1) | 16 620 (41.8) | 0.01 |
| Medications, n (%) | | | | | | |
| Cardiovascular drugs | 80 529 (71.5) | 28 179 (70.9) | 0.01 | 28 054 (71.0) | 28 179 (70.9) | 0.00 |
| Antibiotics | 108 502 (96.3) | 38 405 (96.6) | 0.02 | 38 181 (96.6) | 38 405 (96.6) | 0.00 |
| Oral anticoagulants | 11 111 (9.9) | 4022 (10.1) | 0.01 | 4034 (10.2) | 4022 (10.1) | 0.00 |
| Drugs used in diabetes | 12 935 (11.5) | 4034 (10.1) | 0.04 | 4030 (10.2) | 4034 (10.1) | 0.00 |
| Drugs for obstructive airway diseases | 36 970 (32.8) | 13 118 (33.0) | 0.00 | 13 116 (33.2) | 13 118 (33.0) | 0.00 |
| IBD-specific treatments, n (%) | | | | | | |
| Any IBD-related hospital admissions in the previous year | 564 (1.2) | 714 (6.8) | 0.12 | 584 (5.7) | 714 (6.8) | 0.05 |
| 5-ASA/sulfasalazine | 12 596 (26.8) | 2845 (27.1) | 0.00 | 2857 (27.8) | 2845 (27.1) | 0.01 |
| Budesonide | 646 (1.4) | 272 (2.6) | 0.02 | 277 (2.7) | 272 (2.6) | 0.01 |
| IBD-related procedures | 18 553 (39.5) | 4551 (43.4) | 0.05 | 4473 (43.5) | 4551 (43.4) | 0.00 |
| Endoscopy of the gastrointestinal tract | 7454 (15.9) | 3702 (35.3) | 0.12 | 3543 (34.5) | 3702 (35.3) | 0.02 |
| Arthropathy-specific treatments, n (%) | | | | | | |
| Arthropathy-related procedures | 13932 (31.2) | 8415 (34.7) | 0.09 | 8399 (34.7) | 8415 (34.7) | 0.00 |
| Anti-inflammatory and antirheumatic drugs | 5678 (12.7) | 3462 (14.3) | 0.07 | 3480 (14.4) | 3462 (14.3) | 0.00 |
| Hydroxychloroquine | 540 (1.2) | 845 (3.5) | 0.15 | 801 (3.3) | 845 (3.5) | 0.01 |
| Psoriasis-specific treatments, n (%) | | | | | | |
| 5-ASA/sulfasalazine | 2149 (10.2) | 807 (16.1) | 0.03 | 805 (16.1) | 807 (16.1) | 0.00 |
| Topical corticosteroids | 73 (0.3) | 42 (0.8) | 0.00 | 39 (0.8) | 42 (0.8) | 0.01 |
| Antipsoriatic medication | 7441 (35.4) | 1894 (37.7) | 0.01 | 1900 (37.9) | 1894 (37.7) | 0.00 |
| Topical calcineurin inhibitors | 3731 (17.8) | 1106 (22.0) | 0.00 | 1110 (22.1) | 1106 (22.0) | 0.00 |
| Psoriasis-related procedures | 542 (2.6) | 144 (2.9) | 0.00 | 144 (2.9) | 144 (2.9) | 0.00 |

**Table 1** Continued

| | Baseline IMID cohort | | | Weighted IMID cohort | | |
|---|---|---|---|---|---|---|
| | Unexposed | Exposed | SD | Unexposed | Exposed | SD |
| 5-ASA/sulfasalazine | 2149 (10.2) | 807 (16.1) | 0.03 | 805 (16.1) | 807 (16.1) | 0.00 |

*Total cohort numbers prior to trimming.
NA, not applicable.

of hospitalisation following infection was significantly increased in those ever exposed to systemic corticosteroids in the 120-day period prior to receipt of second vaccination (crude HR: 2.8, 95% CI 1.9 to 4.1; adjusted HR: 2.1, 95% CI 1.4 to 3.2), showing similar results compared with analysis restricting to a ≥7.5 mg daily equivalent dose.

## DISCUSSION

In this large nationwide cohort study of 3 cohorts of patient with IMID, we identified a total of 39 756 immunosuppressive exposed patients matched to 112 629 immunosuppressive unexposed patients with IMID to investigate the risk of COVID-19 infection, hospitalisation and death among patients with IBD, arthropathy and psoriasis following second SARS-CoV-2 vaccination. Meta-analysis of the three cohorts showed an overall 35% increased risk of infection, and 42% increased risk of COVID-19-associated hospitalisation in immunosuppressive exposed compared with immunosuppressive unexposed patients with IMID.

Mortality was not significantly increased in immunosuppressive exposed patients and these events were rare in both groups. Drug class analysis showed that anti-TNF, systemic corticosteroid, and rituximab and other immunosuppressant exposure was significantly associated with both increased risk of COVID-19 infection and hospitalisation following second vaccination in immunosuppressive exposed compared with unexposed patients with IMID.

We attempted to ascertain the effectiveness of SARS-CoV-2 mRNA vaccination among patients with IMID exposed to immunosuppressive therapies, while controlling for the severity of the underlying disease indicating immunosuppressive treatment with a PS model. We found that immunosuppressants were associated with an increased risk of infection, likely due to the impact of immunosuppressive medication on vaccination against COVID-19 infection. This is particularly seen in patients with IBD (HR: 1.6, 95% CI 1.4 to 1.9) but is also present in patients with arthropathy (HR: 1.3, 95% CI 1.1 to 1.4) and, to a lesser extent, in patients with psoriasis (HR: 1.1, 95% CI 0.88 to 1.4). Immunosuppressive exposed patients with psoriasis showed no increased infection risk compared with their unexposed counterparts. Similarly, when assessing risk of hospitalisation following vaccination by immunosuppressive exposure, we find a significantly increased risk in patients with IBD (HR: 2.1, 95% CI 1.0 to 4.1), which is not observed in arthropathy (HR: 1.3, 95% CI 0.9 to 2.0) or psoriasis (HR: 0.6, 95% CI 0.1 to 2.5). The poorer outcomes observed in immunosuppressive exposed patients with IBD are in keeping with wider findings. In a meta-analysis of serological response to SARS-CoV-2 vaccination among IMID-treated patients, patients with IBD were found to have a significantly lower response to first mRNA vaccination dose than patients with rheumatoid arthritis (response rate: 0.49, 95% CI 0.32 to 0.66 and 0.78, 95% CI 0.67 to 0.86, respectively).[22] This may be due to the more extensive disease seen in typical patients with IBD, which often necessitates higher doses of immunosuppressive therapies, over longer periods to achieve disease remission than that required for psoriasis or arthropathy.[23–25] However, the difference

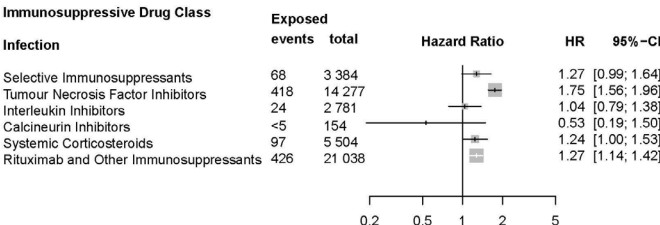

**Figure 3** Risk of infection associated with immunosuppressive drug class exposure (selective immunosuppressants, tumour necrosis factor inhibitors, interleukin inhibitors, calcineurin inhibitors, systemic corticosteroids and other immunosuppressants including rituximab) in patients with immune-mediated inflammatory diseases (IMID).

| | Exposed | | Unexposed | | Hazard Ratio | HR | 95%−CI |
|---|---|---|---|---|---|---|---|
| Infection | events | total | events | total | | | |
| IBD | 287 | 10 477 | 885 | 46 974 | | 1.61 | [1.39; 1.87] |
| Arthropathies | 467 | 24 255 | 800 | 44 650 | | 1.28 | [1.14; 1.43] |
| Psoriasis | 103 | 5 023 | 392 | 20 995 | | 1.10 | [0.88; 1.37] |
| Fixed effect model | 866 | 39 755 | 2 077 | 112 619 | | 1.35 | [1.24; 1.46] |
| Heterogeneity: $I^2 = 79\%$, $\tau^2 = 0.0273$, $p < 0.01$ | | | | | | | |
| Hospitalization | | | | | | | |
| IBD | 15 | 10 477 | 43 | 46 980 | | 2.05 | [1.03; 4.07] |
| Arthropathies | 37 | 24 256 | 59 | 44 650 | | 1.34 | [0.88; 2.03] |
| Psoriasis | <5 | 5 023 | 20 | 20 999 | | 0.59 | [0.14; 2.47] |
| Fixed effect model | <57 | 39 756 | 122 | 112 629 | | 1.42 | [1.01; 2.01] |
| Heterogeneity: $I^2 = 23\%$, $\tau^2 = 0.1737$, $p = 0.27$ | | | | | | | |
| Death | | | | | | | |
| IBD | <5 | 10 477 | 6 | 46 980 | | 0.42 | [0.03; 5.70] |
| Arthropathies | 6 | 24 256 | 13 | 44 650 | | 1.02 | [0.40; 2.61] |
| Psoriasis | 0 | 5 023 | <5 | 20 999 | | 1.00 | [1.00; 1.00] |
| Fixed effect model | <11 | 39 756 | <24 | 112 629 | | 0.92 | [0.38; 2.23] |
| Heterogeneity: $I^2 = 0\%$, $\tau^2 = 0.0647$, $p = 0.53$ | | | | | | | |

0.1 0.5 1 2 10

**Figure 2** Risk of infection, hospitalisation and death associated with immunosuppressive exposure from January 2021 to November 2021 in inflammatory bowel disease (IBD), arthropathy, psoriasis and across all immune-mediated inflammatory diseases (IMID) cohorts.

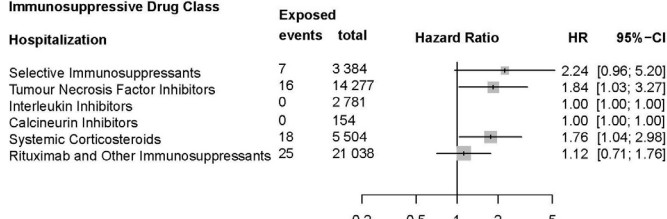

**Figure 4** Risk of hospitalisation associated with immunosuppressive drug class exposure (selective immunosuppressants, tumour necrosis factor inhibitors, interleukin inhibitors, calcineurin inhibitors, systemic corticosteroids, and other immunosuppressants including rituximab) in patients with immune-mediated inflammatory diseases (IMID).

in increased risk of infection and hospitalisation in immunosuppressive exposed IBD compared with unexposed patients with IBD is similar to the other IMID cohorts in this study, with overall pooled IMID cohort meta-analysis showing a significantly increased risk for both these outcomes. These findings indicate a general trend towards poorer outcomes in immunosuppressive exposed patients regardless of IMID cohort.

Reassuringly, immunosuppressant exposure was not associated with increased risk of death due to COVID-19 in any of the cohorts of patient with IMID, suggesting that immunosuppressants do not reduce the effectiveness of vaccination in preventing this important outcome. However, caution should be exercised in the interpretation of this finding as deaths were recorded in either immunosuppressive exposed or unexposed patients and this may be due to the relatively short follow-up period of 11 months in this study. Although Kaplan-Meier plots for risk of infection may appear in contradiction to the overall findings of the primary analysis (with apparent increased rate of COVID-19 infection in the unexposed IMID population in the first 7 months of follow-up), the findings from post-hoc Cox regression analysis by calendar period show that the difference between the groups, reflected in the overall HR, only becomes apparent in the final 3 months of follow-up as the majority of cases of COVID-19 are seen in this period.

Meta-analysis showed an overall increased risk of COVID-19 infection among exposed patients with IMID during the 0–3 and 3–6 months period only. Only exposed patients with arthropathy showed a significantly increased risk of infection in the 3–6 months period. Although this might be interpreted as waning immunity, it is important to note that half of the baseline IMID cohort were censored, largely due to receipt of third vaccination; therefore, the remaining population likely differed substantially from the initial cohort. Hence, this interpretation of period-specific risk estimates should be made tentatively.[26] Although the three periods are not directly comparable, it is likely that our observation of an increased risk of infection in exposed patients in the 0–3 and 3–6 months period reflects a true risk, as is observed in the risk identified over the total follow-up period.

Treatment with TNF-alpha inhibitors, systemic corticosteroid and rituximab and other immunosuppressants was associated with a significantly increased risk of infection, and TNF-alpha inhibitors and systemic corticosteroids were associated with a significantly increased risk of hospitalisation following receipt of second vaccination. This is consistent with previous studies, which suggests that treatment with cytokine inhibitors or B-cell depleting immunosuppressives is related to particularly poor COVID-19 outcomes[27–29]; however, the association with TNF-alpha inhibitors is novel. Our findings of increased risk of infection and hospitalisation, but not death, in sensitivity analysis among patients with IMID exposed to systemic corticosteroids are also in keeping with those other studies of unvaccinated IMID cohorts in Denmark[4] and internationally.[6 30] These findings indicate that corticosteroid exposure weakens the protection conferred by vaccination. As glucocorticoids are known to inhibit the breadth of the immune response, including aspects of both the cellular and humoral immunity induced by mRNA vaccination, these findings appear to be intuitive.[31] The interaction of IMID, the impact of treatments to control disease and response to vaccination, particularly considering the effects of dose and duration of administration is however complex.[32 33] Further studies directly exploring the effects of vaccination while controlling for disease severity and exposure of immunosuppressive drugs by dose and duration would be required to disentangle the association of the different immunosuppressive drug classes with COVID-19 outcomes following vaccination. Such studies would also better inform guidance relating to timelines for SARS-CoV-2 vaccination in relation to the administration of immunosuppressive therapies in patients with IMID.

One of the key strengths of this study is that it is large and population representative, exploring the effectiveness of COVID-19 infection using real-world data from comprehensive, nationwide health registries. Vaccination does not directly correlate with protection from infection and the findings from this work provide important evidence on effectiveness of postmarketing mRNA vaccination in a vulnerable patient group. To our knowledge, this is the first study to assess the effectiveness of SARS-CoV-2 vaccination against COVID-19 infection, hospitalisation and mortality among patients with IMID, based on immunosuppressive exposure. Additionally, our use of PS-weighted regression models allows us to accurately control for the underlying treatment indicating disease, so we are better able to extrapolate the effects of the drug exposure from the disease itself.

Limitations include not being able to extrapolate in the context of the omicron variant, or subsequent subvariants as we restricted to a period of the pandemic where the delta variant was the dominant circulating strain of COVID-19 to ensure consistency in the assessment of our outcomes. Although it is difficult to define a reliable threshold for which we consider a patient unexposed to immunosuppressive medication, it is reassuring that only

approximately 12% of our unexposed group had filled a prescription in the 365 days prior to the 120-day exposure window assessed, and that removing the minimum threshold of 7.5 mg of systemic corticosteroids as an exposure requirement did not change our findings. A lack of individual-level data relating to confounders such as smoking behaviour, risk of occupational exposure to COVID-19, socioeconomic status and dose of drug therapies could potentially limit our findings. Due to lack of availability of such data, we could not account for shielding behaviour in this analysis. There may also be a residual effect of confounding due to unmeasured disease severity not completely accounted for in our PS model. However, these are unlikely to systemically impact the direction of association or strength of significance identified in the risk of infection due to immunosuppressive exposure observed here. We limited our study to IBD, inflammatory arthropathy and psoriasis although other IMID exist, because these are commonly treated with immunosuppressives such as anti-TNF.

In conclusion, our findings suggest a reduced effectiveness of mRNA SARS-CoV-2 vaccination against COVID-19 infection and hospitalisation in patients with IMID receiving immunosuppressive therapies. This risk is particularly seen in patients with IBD and arthropathy, and COVID-19 infection is associated with anti-TNF, systemic corticosteroids, and rituximab and other immunosuppressant exposure, while TNF-alpha inhibitors and systemic corticosteroids were associated with a significantly increased risk of hospitalisation.

**Contributors** RE, DW, AP and TJ developed the study protocol. MTE undertook primary data analysis with support from GP. DW and RE were responsible for first draft of the manuscript. All authors were responsible for interpretation of results and critical revisions to the final manuscript. RE acts as guarantor for the article. The guarantor accepts full responsibility for the work and/or the conduct of the study, had access to the data, and controlled the decision to publish.

**Funding** This work was funded by grants from the Novo Nordisk Foundation (NNF21OC0068631) and the Danish National Research Foundation (DNRF-148).

**Competing interests** None declared.

**Patient and public involvement** Patients and/or the public were not involved in the design, or conduct, or reporting, or dissemination plans of this research.

**Patient consent for publication** Not applicable.

**Ethics approval** Not applicable.

**Provenance and peer review** Not commissioned; externally peer reviewed.

**Data availability statement** Data may be obtained from a third party and are not publicly available. The study was based on data from the Danish National Health registers (https://sundhedsdatastyrelsen.dk). The register data are protected by the Danish Act on Processing of Personal Data and are accessed through application to and approval from the Danish Data Protection Agency and the Danish Health Data Authority. The code is available promptly on request made to the corresponding author.

**ORCID iDs**
Rahma Elmahdi http://orcid.org/0000-0002-7013-9548
Daniel Ward http://orcid.org/0000-0003-0405-7250
Gry Poulsen http://orcid.org/0000-0003-2744-2469
Jesper Hallas http://orcid.org/0000-0002-8097-8708
Anton Pottegård http://orcid.org/0000-0001-9314-5679
Tine Jess http://orcid.org/0000-0002-4391-7332

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
