## [Reviewer comments · BMJ Open]

ARTICLE DETAILS

TITLE (PROVISIONAL)	The Impact of Immunosuppressive Therapy on SARS-CoV-2 mRNA Vaccine Effectiveness in Patients with Immune-mediated Inflammatory Diseases: A Danish Nationwide Cohort Study
AUTHORS	Elmahdi, Rahma; Ward, Daniel; Ernst, Martin; Poulsen, Gry; Hallas, Jesper; Pottegard, Anton; Jess, Tine

VERSION 1 – REVIEW

REVIEWER	Widhani, Alvina Rumah Sakit Umum Pusat Nasional Dr Cipto Mangunkusumo, Department of Internal Medicine Faculty of Medicine Universitas Indonesia
REVIEW RETURNED	18-Jul-2023

GENERAL COMMENTS	COMMENTS TO THE AUTHOR Is the abstract accurate, balanced and complete? Comments: - I suggest to add brief information on vaccine platform (mRNA vaccine), source of cohort data, and disease specific cohort (inflammatory bowel disease, inflammatory arthropathy and psoriasis [ ] not all IMID included) in methods- I also suggest to add the number of patients analyzed in each group in results part Are the methods described sufficiently to allow the study to be repeated? Comments: - I suggest to add exclusion criteria in Methods.- It's better to elaborate the diagnosis of inflammatory arthropathy included, not only give the ICD10.- No information how missing data were addressed- Add information about ethical clearance
---

	Are research ethics (e.g. participant consent, ethics approval) addressed appropriately? Comments:  - No information about ethical clearance. Are they presented clearly? Comments:  - Need clarification for this statement: In results: "A total of 184,391 patients diagnosed with IBD, arthropathy or psoriasis were identified" □ In figure 1: 184,346 IMID patients in the DANISH National Patient Registry ever tested for COVID  - Need clarification on figure 2, supplementary table 4 and this statement ("39,765 IMID patients received immunosuppressive treatment (10,480 IBD, 24,261 arthropathy, and 5,024 psoriasis), and 112,675 IMID patients (47,001 IBD, 44,669 arthropathy, and 21,005 psoriasis) patients did not") Figure 2: Total exposed in infection outcome (39,765) doesn't match with sum for each total IBD, arthropathies and psoriasis (it should be 39,755). In supplementary table 4 it is written that the total exposed in infection is 39,755. I have put circle on the parts that need to be clarified between figure 2 and supp table 4  - In figure 1, I suggest to add how many subjects that were censored and finally analyzed in both exposed and unexposed group - I suggested to add sentences about the result on hospitalization outcome that showed a significantly increased risk of hospitalization in IBD patients, which is not observed in arthropathy or psoriasis. - No statement in the body of manuscript that mentioned supplementary table 3, 4 and 5 - Need clarification for this: "Sensitivity analysis assessing outcomes following any systemic corticosteroid exposure in the 120-day period prior to the index date showed no significant difference in risk of infection (crude HR: 1.2, 95% CI: 1.0, 1.4; 212 adjusted HR: 1.1, 95% CI: 0.95, 1.3) or death (crude HR: 3.1, 95% CI: 1.4, 7.0; adjusted HR: 1.9, 95% CI: 213 0.77, 4.7) (Supplementary Table 6) However, risk of hospitalization following infection was significantly increased in those ever exposed to systemic corticosteroids in the 120-day period prior to receipt of second vaccination (crude HR: 2.8, 95% CI: 1.9, 4.1; adjusted HR: 2.1, 95% CI: 1.4, 3.2)." □ Is this results are shown in supplementary table 6?
--	--

	- No information on number of participants with missing data for each variable of interest Are the discussion and conclusions justified by the results Comments: -Need clarification for this statement: In discussion: “we identified a total of 39,755 immunosuppressive exposed patients matched to 112,619 immunosuppressive unexposed IMID patients to investigate the risk of COVID-19 infection, hospitalization, and death among IBD, arthropathy and psoriasis patients following second SARS-CoV-2 vaccination” □ In figure 1, the number are 39,765 and 112,675. And even though it is subtracted by the number of censored patients (9 exposed and 51 unexposed IMID patients are censored), still it doesn't match. - This statement not entirely correct: “Treatment with TNF-alpha inhibitors, systemic corticosteroid, and rituximab and other immunosuppressants was associated with a significantly increased risk of infection and hospitalization following receipt of second vaccination” □ For hospitalization only Anti-TNF and systemic corticosteroid were associated with an increased risk Are the study limitations discussed adequately? Comments: Another limitation that should be addressed that this cohorts only on IBD, psoriasis and inflammatory arthropathy. There is another IMIDs that are not covered in the cohort.
--	--

REVIEWER	Motedayyen, Hossein Kashan University of Medical Sciences
REVIEW RETURNED	21-Jul-2023

GENERAL COMMENTS	 1. Suggest a revision, mainly for its language. It is important that the message being conveyed in the manuscript is as unambiguous as possible. 2. What is the novelty of this study? While there are many reports revealing immunosuppressive drugs have a negative impact on immune responses in those treated with these agents and enhance susceptibility to various infections. 3. Regarding the fact that the chance of exposure to COVID-19 differs among different persons, I suggest that the effects of COVID-19 vaccination in IMID patients could be investigated to clarify whether COVID-19 vaccines provide an acceptable level of immunity against the disease. 4. How did the researchers control the impacts of confounders such as, the chance of occupational exposure to COVID-19, quality of life style, and other factors affecting the immune system, in two groups of the study?
---

	5. Criteria and clinical guidelines used to diagnose IMID should be mentioned or cited in the methods section. 6. Please use the full names (instead of acronyms) For the EU/EAA.
--	--

REVIEWER	Escarela, Gabriel Universidad Autonoma Metropolitana, Departamento de Matemáticas
REVIEW RETURNED	19-Nov-2023

GENERAL COMMENTS	The methods employed assume that the events will eventually occur. It is clear there is a cured fraction, and it is not possible to assess if there is sufficient follow-up to use cure models. No Kaplan-Meier plots are shown. Also, when death is studied in the presence of a competing cause, semi-competing risks methods should be implemented.
--

VERSION 1 – AUTHOR RESPONSE

Reviewer 1:

Comments:

- I suggest adding brief information on vaccine platform (mRNA vaccine), source of cohort data, and disease specific cohort (inflammatory bowel disease, inflammatory arthropathy and psoriasis◊ not all IMID included) in methods.

Thank you for this comment. We establish the vaccine type used (mRNA platform) in the Materials and Methods section, line 142 “SARS-CoV-2 mRNA (Pfizer-BioNTech or Moderna)”.

Our cohort data comes from nationwide registers, as described in Data Sources, line 134-144. Additionally, further details on the registers are available in the referenced articles (Pottegård A et al., Clinical epidemiology. 2020; Voldstedlund M et al., Euro surveillance bulletin 2014; Schmidt M et al., Clinical epidemiology. 2015; Pottegård A, et al., International journal of epidemiology. 2017). We have now clarified the disease specific cohorts, listing the disease included in each, line 150-153.

- I also suggest to add the number of patients analyzed in each group in results part

Thank you for offering the opportunity to clarify this. We detail the number of patients analysed in each group on line 244-246, “39,765 IMID patients received immunosuppressive treatment (10,480 IBD, 24,261 arthropathy, and 5,024 psoriasis), and 112,675 IMID patients (47,001 IBD, 44,669 arthropathy, and 21,005 psoriasis) patients did not.” These numbers are described in the flow chart in Figure 2.

• Are the methods described sufficiently to allow the study to be repeated?

Comments:

- I suggest to add exclusion criteria in Methods.

Thank you for this comment. The text is now revised to clarify the exclusion criteria, line 141-142, “Exclusion criteria were: not receiving two doses of SARS-CoV-2 mRNA vaccine and migration prior to receipt of second vaccination.”

- It's better to elaborate the diagnosis of inflammatory arthropathy included, not only give the ICD10.

Thank you for this comment. As mentioned above, we have now clarified the disease specific cohorts, listing the disease included in each, line 161-162.

- No information how missing data were addressed

Thank you for the opportunity to clarify this. The methods are now revised to clarify in line 170-172, "As the registers are complete for the presence of patients up to emigration or death, therefore all patients are retained until the event and there is no missing data."

- Add information about ethical clearance

Thank you for the opportunity to clarify this. We now explain in line 144-145, "Ethics board review is not required for epidemiological research using nationwide registers in Denmark as data is pseudonymised and does not involve patients." More information regarding Danish National Health registry data is also included in the Data Sharing Statement (line 442-446).

• Are research ethics (e.g. participant consent, ethics approval) addressed appropriately?

Comments:

- No information about ethical clearance.

Thank you, for bringing this to our attention. It is now corrected. Please see the previous comment regarding ethical approval and participant consent for completion.

• Are they presented clearly?

Comments:

- Need clarification for this statement:

o In results: "A total of 184,391 patients diagnosed with IBD, arthropathy or psoriasis were identified"◊ In figure 1: 184,346 IMID patients in the DANISH National Patient Registry ever tested for COVID

We thank the reviewer for bringing this to our attention and apologise for the oversight. The corrected (and corresponding) number is now updated in the Figure 1. "184,346 IMID patients in the Danish National Patient Registry ever tested for COVID."

- Need clarification on figure 2, supplementary table 4 and this statement ("39,765 IMID patients received immunosuppressive treatment (10,480 IBD, 24,261 arthropathy, and 5,024 psoriasis), and 112,675 IMID patients (47,001 IBD, 44,669 arthropathy, and 21,005 psoriasis) patients did not")

Figure 2: Total exposed in infection outcome (39,765) doesn't match with sum for each total IBD, arthropathies and psoriasis (it should be 39,755). In supplementary table 4 it is written that the total exposed in infection is 39,755.

I have put circle on the parts that need to be clarified between figure 2 and supp table 4

We thank the reviewer for bringing this to our attention and apologise for the oversight. Please see the corrected number for total IMID included for COVID infection analysis in Figure 1 as 39,755.

Please also see deletion of repeated entry in COVID hospitalization for combined cohorts for total exposed and unexposed, corrected numbers of total exposed patients with arthropathy for

hospitalization and death as 24,256, and finally corrected numbers for total unexposed group included in the secondary analyses, by IMID type in Supplementary Table 4.

Finally, we have corrected the number removed due to censoring in the manuscript main text body (line 246). Unfortunately, we are not able to expressly specify which IMID these patients were subsequently included into the secondary analysis for (i.e., IBD, psoriasis or arthropathy) as these individuals are fewer than five in total and this would be a violation of anonymity requirements.

- In figure 1, I suggest to add how many subjects that were censored and finally analyzed in both exposed and unexposed group

Thank you for this excellent suggestion and opportunity to provide clarity in our final cohort for inclusion and analysis. We have now updated Figure 1. to reflect the censoring and those included in primary and secondary analyses.

- I suggested to add sentences about the result on hospitalization outcome that showed a significantly increased risk of hospitalization in IBD patients, which is not observed in arthropathy or psoriasis.

We thank the reviewer for highlighting this important result. We now explicitly refer to this finding, qualifying the contribution of IBD patients to the overall increased risk in hospitalization among IMID patients (line 269-271).

- No statement in the body of manuscript that mentioned supplementary table 3, 4 and 5

We thank the reviewer for bringing this to our attention. We have now included a reference to Supplementary Tables 3 (line 184), 4 (line 251) and 5 (line 281).

- Need clarification for this: "Sensitivity analysis assessing outcomes following any Systemic corticosteroid exposure in the 120-day period prior to the index date showed no significant difference in risk of infection (crude HR: 1.2, 95% CI: 1.0, 1.4; 212 adjusted HR: 1.1, 95% CI: 0.95, 1.3) or death (crude HR: 3.1, 95% CI: 1.4, 7.0; adjusted HR: 1.9, 95% CI: 213 0.77, 4.7) (Supplementary Table 6) However, risk of hospitalization following infection was significantly increased in those ever exposed to systemic corticosteroids in the 120-day period prior to receipt of second vaccination (crude HR: 2.8, 95% CI: 1.9, 4.1; adjusted HR: 2.1, 95% CI: 1.4, 3.2)."

◇ Is this results are shown in supplementary table 6?

We thank the reviewer for bringing this point to our attention. These are indeed the findings from the sensitivity analysis, for which the criteria to meet a minimum threshold for immunosuppressive exposure equivalent to corticosteroid use of 7.5 mg daily in the 120-day period prior to vaccination, is removed. Supplementary Table 6 presents the analysis for risk of infection, hospitalization or death associated with immunosuppressive exposure in the 120 days preceding vaccination, not any immunosuppressive exposure. Reference to Supplementary Table 6 has now been moved accordingly.

- No information on number of participants with missing data for each variable of interest

Due to complete registries, there is no missing data in relation to variables such as diagnoses and dispensed medications. We clarify this now in lines 170-172, "As the registers are complete for the

presence of patients up to emigration or death, therefore all patients are retained until the event and there is no missing data.

- Are the discussion and conclusions justified by the results.

Comments:

-Need clarification for this statement:

In discussion: “we identified a total of 39,755 immunosuppressive exposed patients matched to 112,619 immunosuppressive unexposed IMID patients to investigate the risk of COVID-19 infection, hospitalization, and death among IBD, arthropathy and psoriasis.

patients following second SARS-CoV-2 vaccination”◇ In figure 1, the number are 39,765 and 112,675. And even though it is subtracted by the number of censored patients (9 exposed and 51 unexposed IMID patients are censored), still it doesn't match.

We thank the reviewer for highlighting this unclear phrasing. We have now clarified that the numbers for the total patients included in primary analysis (i.e., to assess risk of COVID infection following vaccination) and total numbers included in secondary analyses (for risk of hospitalization or death following COVID infection) in line 316-317.

Eleven patients in the exposed group and fifty-one patients in the unexposed group are censored from primary analysis (due to emigration or infection on the date of inclusion) and 10 patients in the exposed group and 46 patients in the unexposed group are also censored (only due to emigration) for the purposes of the secondary analyses, where receiving a positive test on the index date does not preclude from inclusion. These numbers have been corrected and are now consistent in Figure 2., Supplementary Table 4., and in Results (line 240-251), as per our response to prior comment regarding number of patients included in each cohort.

- This statement not entirely correct: “Treatment with TNF-alpha inhibitors, systemic corticosteroid, and rituximab and other immunosuppressants was associated with a significantly increased risk of infection and hospitalization following receipt of second vaccination”.

◇ For hospitalization only Anti-TNF and systemic corticosteroid were associated with an increased risk.

Thank you for pointing out this oversight, it is now corrected to read in lines 376-379, “Treatment with TNF-alpha inhibitors, systemic corticosteroid, and rituximab other immunosuppressants was associated with a significantly increased risk of infection, and TNF-alpha inhibitors and systemic corticosteroids were associated with a significantly increased risk of hospitalization following receipt of second vaccination.”

- Are the study limitations discussed adequately?

Comments:

Another limitation that should be addressed that this cohorts only on IBD, psoriasis and inflammatory arthropathy. There is another IMIDs that are not covered in the cohort.

Thank you for mentioning this limitation, which we have now clarified in line 420-422, “We limited our study to IBD, inflammatory arthropathy, and psoriasis although other IMIDs exist, because these are commonly treated with immunosuppressives such as anti-TNF.”

Reviewer 2:

Comments to the Author:

1. Suggest a revision, mainly for its language. It is important that the message being conveyed in the manuscript is as unambiguous as possible.

Thank you for suggesting clarification of the message. We have now revised the conclusion to state this message unambiguously, lines 426-429, “In conclusion, our findings suggest a reduced effectiveness of mRNA SARS-CoV-2 vaccination against COVID-19 infection and hospitalization in

IMID patients receiving immunosuppressive therapies. This risk is particularly seen in IBD and arthropathy patients, and COVID-19 infection is associated with anti-TNF, systemic corticosteroids, and rituximab and other immunosuppressant exposure, while TNF-alpha inhibitors and systemic corticosteroids were associated with a significantly increased risk of hospitalization.”

2. What is the novelty of this study? While there are many reports revealing immunosuppressive drugs have a negative impact on immune responses in those treated with these agents and enhance susceptibility to various infections.

Thank you for this comment. As we mention in lines 401-403, this study is to our knowledge the first to study the effect of immunosuppressives on SARS-CoV-2 vaccination in patients with IMIDs. Furthermore, this study has several strengths as it is based on a large and population-representative cohort.

3. Regarding the fact that the chance of exposure to COVID-19 differs among different persons, I suggest that the effects of COVID-19 vaccination in IMID patients could be investigated to clarify whether COVID-19 vaccines provide an acceptable level of immunity against the disease.

Thank you for this comment regarding COVID-19 vaccine effectiveness. While this is relevant, it addresses a different question to that which we sought to investigate, in that it refers to the effectiveness of vaccination in IMID patients compared to unvaccinated IMID patients. In contrast, we have investigated the effect of immunosuppressants on the risk of COVID-19 outcomes in vaccinated patients with IMIDs.

4. How did the researchers control the impacts of confounders such as, the chance of occupational exposure to COVID-19, quality of lifestyle, and other factors affecting the immune system, in two groups of the study?

Thank you for this comment regarding confounders. We have mentioned a number of unmeasured confounders in the limitations section in lines 407-422, and now include mention of occupational exposure to COVID-19. The confounders with relevance to effects on the immune system that we included were comorbidities and co-medications. We also included the frequency of testing in the month prior to the index date to control for bias that could relate to virus prevalence and risk of exposure.

5. Criteria and clinical guidelines used to diagnose IMID should be mentioned or cited in the methods section.

Thank you for the opportunity to clarify this. We have now included mention of clinical guidelines and citations for IBD and rheumatoid arthritis in the methods section, lines 161-162, “Registration of IMIDs is based on clinical diagnoses, in line with national and international guidelines, such as ECCO-ESGAR guidelines for IBD diagnosis and ACR/EULAR guidelines for rheumatoid arthritis (17,18).”

6. Please use the full names (instead of acronyms) For the EU/EAA.

Thank you, this is now corrected in the text, line 92.

Reviewer 3:

Comments to the Author:

The methods employed assume that the events will eventually occur. It is clear there is a cured fraction, and it is not possible to assess if there is sufficient follow-up to use cure models. No Kaplan-Meier plots are shown. Also, when death is studied in the presence of a competing cause, semi-competing risks methods should be implemented.

We thank the reviewer for this thoughtful comment. While it is true that the Cox regression model theoretically assumes that all individuals would eventually get the outcome if there were enough follow-up time and no censoring, the model is commonly used for problems where few individuals will get the outcome within their lifetime, as long as all individuals can potentially get the outcome. In the

primary analysis, where the outcome is COVID-19 infection, all individuals can potentially get the outcome – and the vast majority did in fact get COVID-19 in the months after end of follow-up (by spring 2022 estimated 70-80% of Danish population had had a COVID-19 infection).

We are uncertain what the cure fraction would mean in current study. Usually cure fractions are estimated in studies where you can distinguish between short- and long-term survival, but it is not clear to us how this would apply to risk of COVID-19 infection.

Semi-competing risk models might help to disentangle the interplay between COVID hospitalization and death after a positive COVID-19 test but given that COVID death is a secondary outcome and very few COVID deaths were observed in the study period, we think it is outside the scope of the paper to include a semi-competing risks model. However, since COVID death can be seen as a competing risk to getting a COVID hospitalization, we have performed a sensitivity analysis for COVID hospitalization using a Fine-Gray model including COVID death as competing risk. This sensitivity analysis did not change the results and we have therefore not included it in the manuscript.

Please see the addition of Kaplan-Meier plots for probability of infection over the calendar year (January-November 2021) from index date as Supplementary Figure 1 and additionally a post-hoc analysis of hazards for risk of infection between immunosuppressive exposed and unexposed group broken down by calendar time (Jan-April, May-Aug and Sept-Nov 2021) as Supplementary Table 7. This is now also referenced in the Results (line 291-295). Finally, we discuss the increased hazards of infection in the immunosuppressed group, which is seen exclusively in the final three months of follow-up (most likely due to changes in public guidelines, isolation strategies and testing strategy) in the Discussion (line 359-364). We additionally add a sentence under limitations, explaining the limitation of lack of shielding data in the complete interpretation of post-hoc analysis results (line 416-417).

VERSION 2 – REVIEW

REVIEWER	Widhani, Alvina Rumah Sakit Umum Pusat Nasional Dr Cipto Mangunkusumo, Department of Internal Medicine Faculty of Medicine Universitas Indonesia
REVIEW RETURNED	05-Jan-2024

GENERAL COMMENT S	REVIEW Thank you for the revisions that have addressed previous suggestions. I still have some questions that need to be clarified.  • Can you give clarification about figure 1 and supplementary table 4? Is the data in circle in figure 1 and supplementary table 4 should be the same? I think the one in the supplementary table 4 is the right one? Because the sum in figure one is not match (10480+24261+5024 is not 39756 and 47001+44669+21005 is not 112629)
---

Figure 1.

- Is the number of events and total for each outcome/exposure/autoimmune disease in figure 2 and supplementary figure 4 should be the same? Because there are some data that are similar, but the others are different.

27
28
29
30
31
32
33
34
35
36
37
38
39
40
41
42

COVID-19 Hospitalization*										
IBD	10 477	15	2.8 (1.7-4.7)	5 280	46 980	43	2.1 (1.5-2.8)	20 892	2.8 (1.5-5.1)	2.1 (1.0-4.1)
Arthropathy	24 256	37	3.1 (2.3-4.3)	11 813	44 650	59	2.8 (2.2-3.6)	21 215	1.3 (0.9-2.0)	1.3 (0.9-2.0)
Psoriasis	5 023	<5	-	2 284	20 999	20	2.1 (1.4-3.3)	9 479	0.5 (0.1-2.3)	0.6 (0.1-2.5)
Combined cohorts	39 756	52	-	19 437	112 629	122	-	51 594	1.6	1.4

Figure 2.

Supplementary Table 4. Risk of infection, hospitalisation and death associated with immunosuppressive therapy in vaccinated patients with immune-mediated inflammatory diseases.

	Exposed				Unexposed				Crude HR (95% CI) ***	Weighted HR (95% CI) ***
	Total	Events	Incidence rate (95% CI) **	PY	Total	Events	Incidence rate (95% CI) **	PY		
COVID-19 Infection*										
IBD	10 477	287	54.7 (48.7-61.4)	5 246	46 974	885	42.5 (39.8-45.4)	20 809	1.7 (1.5-1.9)	1.6 (1.4-1.9)
Arthropathy	24 255	476	40.3 (36.8-44.1)	11 822	44 650	800	37.8 (35.3-40.6)	21 140	1.3 (1.1-1.4)	1.3 (1.1-1.4)
Psoriasis	5 023	103	45.3 (37.4-55.00)	2 273	20 995	392	41.5 (37.6-45.8)	9 441	1.1 (0.9-1.4)	1.1 (0.9-1.4)
Combined cohorts	39 755	866	-	19 341	112 619	2 077	-	51 390	1.4 (1.3-1.5)	1.4 (1.2-1.5)
COVID-19 Hospitalization*										
IBD	10 477	15	2.8 (1.7-4.7)	5 280	46 980	43	2.1 (1.5-2.8)	20 892	2.8 (1.5-5.1)	2.1 (1.0-4.1)
Arthropathy	24 256	37	3.1 (2.3-4.3)	11 873	44 650	59	2.8 (2.2-3.6)	21 215	1.3 (0.9-2.0)	1.3 (0.9-2.0)
Psoriasis	5 023	n=5	-	2 284	20 999	20	2.1 (1.4-3.3)	9 479	0.5 (0.1-2.3)	0.6 (0.1-2.5)
Combined cohorts	39 756	52	-	19 437	112 629	122	-	51 594	1.6	1.4

For peer review only - <http://bmjopen.bmj.com/site/about/guidelines.xhtml>

of 83

BMJ Open

									(1.1-2.2)	(1.0-2.0)
Death*										
IBD	10 477	n=5	-	5 281	46 980	6	0.3 (0.1-0.6)	20 897	1.1 (0.1-9.7)	0.4 (0-5.6)
Arthropathy	24 256	6	0.51 (0.2-1.1)	11 875	44 650	13	0.6 (0.4-1.1)	21 222	1.0 (0.4-2.7)	1.0 (0.4-2.6)
Psoriasis	5 023	0	-	2 284	20 999	n=5	-	9 480	-	-
Combined cohorts	39 756	6	-	19 440	112 629	19	-	51 599	1.0 (0.4-2.5)	0.9 (0.4-2.2)

- Need clarification for this statement:
One arthropathy patient in the exposed group, and 5 IBD or psoriasis patients in the unexposed group are excluded from the infection analysis
- Referring to supplementary table 4, is this should be: One arthropathy patient in the exposed group, and 10 IBD or psoriasis patients in the unexposed group are excluded from the infection analysis

VERSION 2 – AUTHOR RESPONSE

Reviewer Report:

Reviewer: 1

Dr. Alvina Widhani, Rumah Sakit Umum Pusat Nasional Dr Cipto Mangunkusumo

Comments to the Author:

Thank you for the revisions that have addressed previous suggestions. I still have some questions that need to be clarified. I have attached the reviews below.

Can you give clarification about figure 1 and supplementary table 4?

Is the data in circle in figure 1 and supplementary table 4 should be the same? I think the one in the supplementary table 4 is the right one? Because the sum in figure one is not match (10480+24261+5024 is not 39756 and 47001+44669+21005 is not 112629).

We thank you for the highlighting this mistake. Please see corrected final figure in agreement with supplementary table 4. We have also corrected these so the main manuscript is also in agreement with these numbers.

Is the number of events and total for each outcome/exposure/autoimmune disease in figure 2 and supplementary figure 4 should be the same? Because there are some data that are similar, but the others are different.

We thank you for the highlighting this. Both figure 2 and supplementary table 4 are now in agreement. We would like to bring the reviewers attention to the need to censor categories where patient numbers total less than 5 for the purpose of anonymity. You will now see corrected numbers that do not allow the reader to back-calculate categories occupied by less than 5 patients, as is the cases for number of psoriasis patients with positive secondary outcomes in the analysis.

Need clarification for this statement:

One arthropathy patient in the exposed group, and 5 IBD or psoriasis patients in the unexposed group are excluded from the infection analysis

Referring to supplementary table 4, is this should be: One arthropathy patient in the exposed group, and 10 IBD or psoriasis patients in the unexposed group are excluded from the infection analysis

We thank the reviewer for highlighting this error. Please find this sentence corrected in the main body of the manuscript (line 234-235).